# Fire Spread Characteristics of Metal-Polyethylene Sandwich Panels

Ru Zhou [1], Zhihao Chen [1], Yinke Fan [1], Zhengjiang Yu [1], Jianan Qian [1,*] and Juncheng Jiang [1,2]

1   Jiangsu Key Laboratory of Urban and Industrial Safety, College of Safety Science and Engineering, Nanjing Tech University, Nanjing 211816, China; maxmuse.zhou@njtech.edu.cn (R.Z.); 201961201043@njtech.edu.cn (Z.C.); 201761100036@njtech.edu.cn (Y.F.); 201761100040@njtech.edu.cn (Z.Y.); jcjiang@njtech.edu.cn (J.J.)
2   School Environment & Safety Engineering, Changzhou University, Changzhou 213164, China
*   Correspondence: 2368@njtech.edu.cn

**Abstract:** An experimental study was conducted to determine the characteristics of the flame spread and droplets of metal-polyethylene (PE) sandwich panels during combustion. The mass-loss rate, average flame height, temperature, and fire spread rate were investigated. The results showed that the fire spread rate, mass change of the droplets, average flame height, and temperature increased with an increase in the sample length, except for the mass loss rate of the 40 cm-long sample. The time interval between the droplets decreased, and the flame pulsation frequency increased. The relationship between the flame height and sample length was determined. During the combustion process, bending deformation and top flame phenomena occurred due to the shrinkage of the PE, which increased the fire risk. The distance between the outer surface of the expanded metal aluminum layer and the insulation panel increased with an increase in the panel length. A schematic diagram of the fire spread of the metal sandwich panel was established based on the observations and theoretical analysis. The mechanism and combustion behavior of the metal sandwich panels were determined to provide references for the construction of metal sandwich panels of exterior walls.

**Keywords:** metal-polyethylene sandwich panel; fire spread rate; flame characteristics; bending deformation; top flame

## 1. Introduction

Polymer is a widely used insulation material in industrial buildings, warehouses, and other locations. In recent years, many major fires have been caused by the insulation materials in the exterior walls of buildings worldwide. The facade of an electronics factory in the Busan Industrial Park in South Korea caught on fire in 2012. In the same year, a fire broke out in the workers' dormitory in the Indian-controlled Kashmir region, and at least 10 people died in the collapse due to the poor fire resistance of the PE sandwich board. In 2017, 79 people were killed by fire in a 24-story apartment building (Grenfell Tower) in west London, England [1]. These fire accidents have caused numerous casualties. The building exterior insulation material in high-rise building fire accidents is important to the fire safety, which is compared with fire safety codes to identify key areas for improvement [2,3]. Therefore, it is crucial to study the flame spread characteristics of metal sandwich panels.

In this study, the surface of the metal sandwich panel is an aluminum laminate, and the sandwich layer is polyethylene (PE). Sandwich panels with combustible cores have several fire hazards. They are prone to the delamination of the steel faces due to the decomposition of the resin between the panel and the core and thermal stresses and expansion [4,5]. Pyrolysis gases can travel through the panel to other compartments due to panel distortion and the openings of the joints between the panels [6]. Fire in the core may go unnoticed, and flames can spread through the core [7]. In addition, dense smoke is caused by the pyrolysis of the insulation core, and there is potential for a smoke gas explosion due to the mixture of pyrolysis gases and hot air [8].

Several studies have been conducted on the fire behavior and mechanism of metal sandwich panels. Zhuo found that external radiation had a substantial influence on the fire spread parameters of PE and expanded polystyrene (EPS) metal sandwich panels. It was found that the external radiation intensity of metal panels with PE and EPS sandwich layers was proportional to the flame spread rate and the flame height. When the PE and EPS metal sandwich panels were exposed to the same external radiation, the flame height, flame temperature, and flame spread rate of the EPS metal panel were slightly higher than those of the PE metal panel [9]. Oleszkiewiczl and Burgess tested the fire spread characteristics of different types of insulation materials for metal sandwich panels using a full-scale experimental platform. The fire hazard and fire spread characteristics of the insulation materials were classified according to the fire spread rate, height, fire spread distance, and other parameters. In the combustion of a metal sandwich plate, it is also essential to determine the stability of the fixed plate and the influence on the surrounding combustion [10,11]. Griffin performed an ISO9705 standard test, investigated the fire behavior of sandwich panels (aluminum panel with an EPS core), and determined the influences of the thickness and construction of the core material. The study found that an appropriate core material thickness and construction method prevented the fire spread of the sandwich panels [12]. Jianbo used a self-made combustion device for metal sandwich panels with rock wool and PE as the core material. It was found that the metal sandwich panels with the flame-retardant grades B1 and B2 remained stable during the combustion, limiting the flame spread [13]. Lie used a full-scale experimental platform to analyze the influence of the flame spread rate of EPS metal sandwich panels on the combustion of the insulation around the panels. It was found that the influence on the surrounding insulation material was negligible when the fire spread rate of the EPS sandwich was lower than 18 mm/s [14]. You used different areas covered with gasoline to change the fire source power and analyze the fire spread of polystyrene foam sandwich panels. Thermocouples were inserted into the interior of the board to measure the temperature variation on the inside of the sheet. It was found that a first-order exponential decay function provided the best fit of the temperature at different measuring points in the horizontal and vertical positions. The application of external wall insulation panels had a positive effect [15]. These studies have analyzed the influence of external radiation, the sandwich panel material, sandwich panel thickness, and other parameters on the fire spread.

However, few studies have focused on the length (width) effect of metal sandwich panels. There appears to be little consensus on the effects of the panel length. In most models, it was assumed that the length of the fuel sample was invariable [16–18], and the sample width and thickness were used as experimental parameters [19]. However, the flame spread behavior of panels with different widths and lengths has not been researched. Furthermore, there are few studies on the effects of the sample length on the downward flame spread of metal-PE sandwich panels. Therefore, in this study, metal sandwich panels with a PE core were chosen to determine the combustion characteristics. The fire spread behavior and dripping behavior of the thermoplastic material in the sandwich panel with different lengths were investigated. The influence of the panel length on the flame spread characteristics, including the flame shape, flame spread rate, mass loss rate, flame height, and the internal temperature of the sample, was analyzed.

## 2. Experiment

### 2.1. Experimental Apparatus and Methods

This experimental device was a self-designed combustion device for the metal sandwich panel, as shown in Figure 1. The test device consisted of four parts: the weighing system, the video image acquisition system, the temperature acquisition system, and the ignition source.

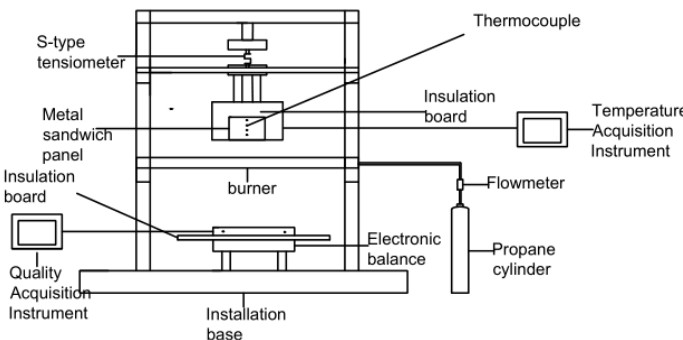

(**a**) Front view of the combustion device

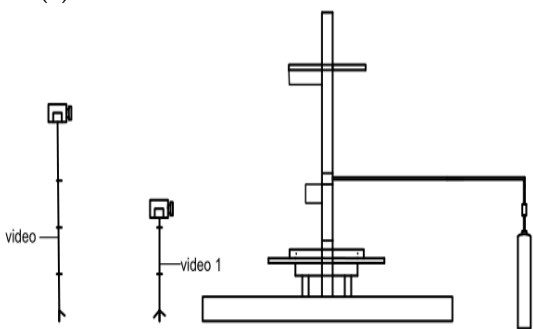

(**b**) Side view of the combustion device

**Figure 1.** Combustion device.

### 2.1.1. Weighing System

The weighing system was divided into two parts. An S-type load cell (SHSIWI, SH-1 K-5 K) was used to measure the mass change of the insulation materials of the metal sandwich panel. The range was 0–20 kg, with an accuracy of 0.1 g. An electronic balance (Sartorius, MSA70201S-000-D0) was used to measure the droplet mass change. The precision was 0.01 g, and the maximum weight was 70.2 kg. The data collection interval was 1 s.

### 2.1.2. Video Image Acquisition System

A digital camera (Panasonic, HC-V180GKC-K) was used with 90× intelligent zoom, 50× optical zoom, and standard illumination of 1400lx. The camera was used to record the fire spread on the surface of the metal sandwich panels during the experiment.

### 2.1.3. Ignition Source

The fire source was 99.9% propane gas. A flow meter was used to measure the power of the fire source [20].

### 2.1.4. Temperature Acquisition System

K-type thermocouples and a data acquisition instrument (Keysight, 34970A) were used to record the temperature change in the middle of the metal sandwich panel during the combustion, as shown in Figure 2. The inspection instrument had 3 channels, and each channel could be connected to 16 thermocouples. Thus, 48 thermocouples were used at the same time. The minimum time interval was 4 ms. There were four positions from top to bottom ($P_1$, $P_2$, $P_3$, and $P_4$). The distance between the points was 5 cm, and the distance between P and the bottom edge was 3 cm. Four bolts were used to attach the metal sandwich panel at the four vertices. The electronic balances, S-type dynamometer, and temperature inspection instruments were connected to the computer to record the data.

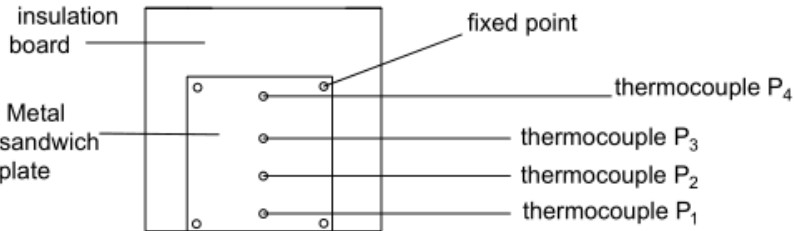

**Figure 2.** Schematic diagram of the thermocouple layout.

### 2.1.5. Metal Sandwich Panel Sample

The test sample (Shanghai JiXiang LTD, JXPE802, Shanghai, China) was purchased from a construction company. The surface was aluminum alloy, and the core was PE. The sample had a thickness of about 4 mm. Five samples with lengths of 20 cm, 25 cm, 30 cm, 35 cm, and 40 cm and a width of 20 cm were fabricated. Each experiment lasted for 30 min, and the tests with the samples of the same size were repeated three times under the same condition to minimize the experimental error. The power of the fire source was set to 5 kW, according to the heating rate used by An [21].

## 3. Results and Discussion

### 3.1. Combustion Behavior of the Metal Sandwich Panels

### 3.1.1. Combustion Phenomena

In high-rise building fires, the PE molten droplets drip down in the vertical direction and initiate the exterior insulation material of the lower floors [22]. Therefore, molten droplets play an important role in the spreading of high-rise building fires. As shown in Figures 3 and 4, according to the firing characteristics and dripping process, the whole combustion process was divided into four stages. After the ignition, the temperature of the PE sandwich panel increased, which was defined as the preheating stage of combustion before the first firing droplet was detected. Subsequently, intermittent droplets were observed. However, the time interval between the appearance of the droplets was unpredictable. This stage was defined as the initial stage of combustion. As the fire continued, droplets appeared continuously. The droplets accumulated in the middle of the sandwich panel until the force of gravity was greater than the viscous force on the metal surface of the sandwich panel. At this time, the droplets broke free from the constraining force of the surface aluminum laminate and dripped into the tray, forming the molten fire. A series of continuous droplets was observed, but the time interval between the different groups of continuous droplets remained the same. A cavity occurred at the bottom of the panel, and the PE on the upper side melted and flowed down after sustained heating. The number of droplets decreased as the heating continued. This stage was defined as the stable stage of combustion. After the stable stage, the droplets were occasionally observed with the number of droplets decreasing and the time interval increasing, and until no droplets were observed, which was considered the burnout stage. The phenomena of the four stages agree with the temperature curves.

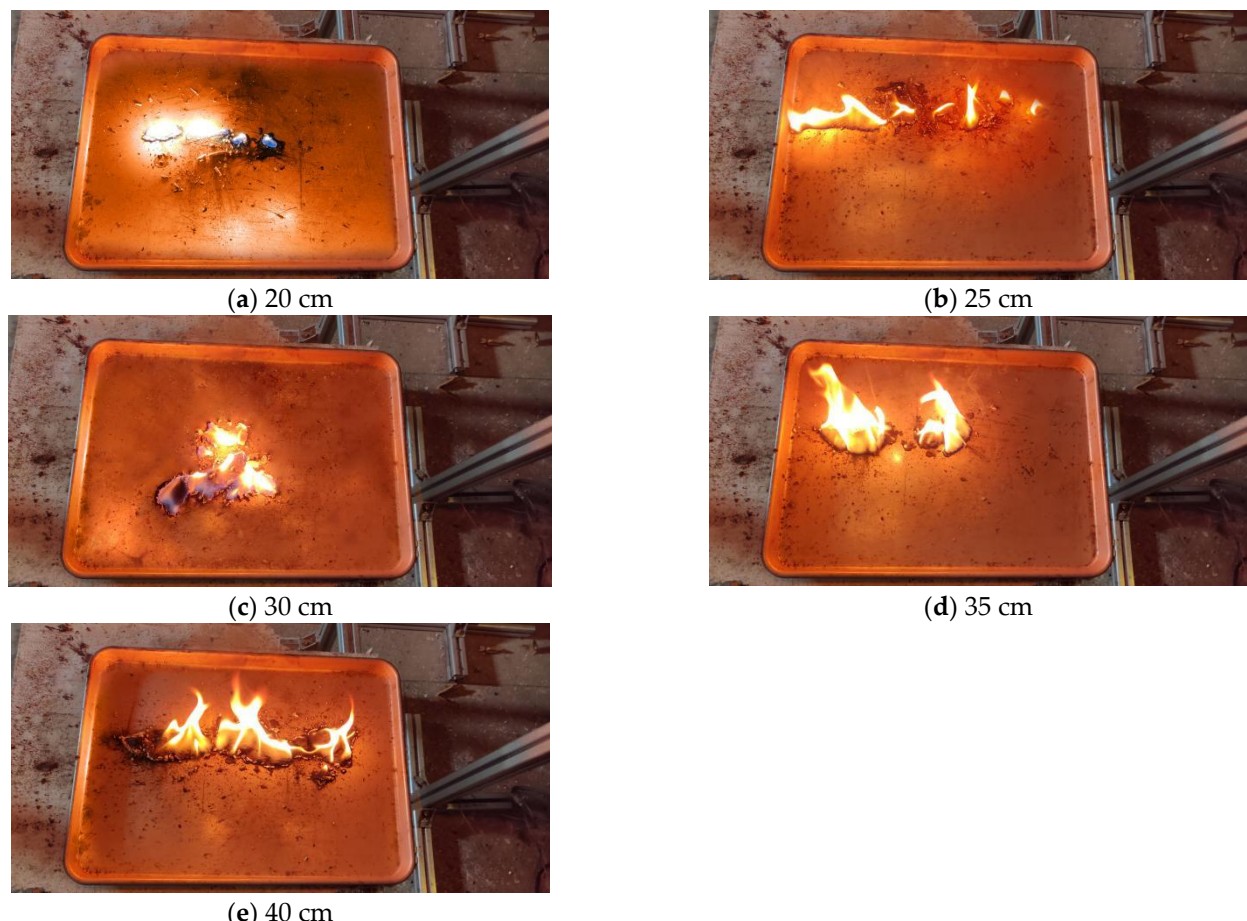

(**a**) 20 cm  (**b**) 25 cm

(**c**) 30 cm  (**d**) 35 cm

(**e**) 40 cm

**Figure 3.** The molten fire in the stable stage of combustion.

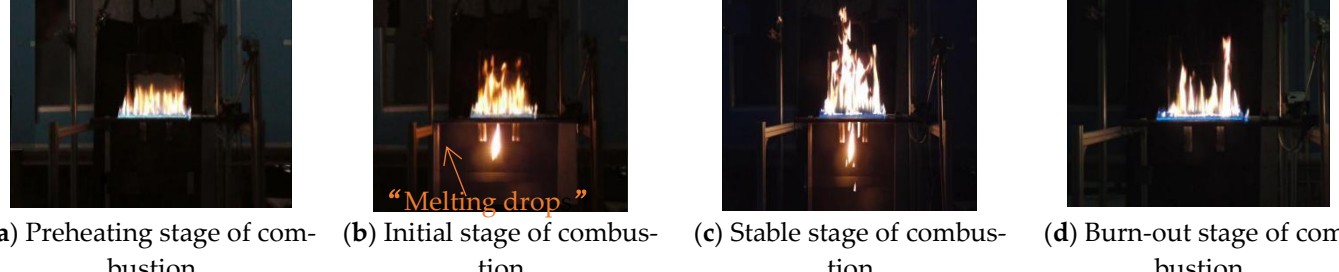

(**a**) Preheating stage of combustion  (**b**) Initial stage of combustion  (**c**) Stable stage of combustion  (**d**) Burn-out stage of combustion

**Figure 4.** Flame shape in the different combustion stages for a panel length of 35 cm.

In the experiments, the flame shape changed and was unstable in different stages. The flame shapes in the stable combustion stage, which occurred nearly 60 s after the drops were observed, were different for the different panel lengths, as shown in Figure 5.

The schematic diagram of the vertical fire spread of the metal sandwich panel was established by improving the physical combustion model [23,24], as shown in Figure 6. The burning of the fire source released a substantial amount of heat, and heat conduction, heat convection, and thermal radiation resulted in the heat transfer to the unburned PE. The PE was heated and expanded, but the aluminum surface constrained the flow of the molten PE. The combination of gravity, channel constraints, and viscosity made the molten PE flow toward the bottom of the panel. The molten PE closest to the flame decomposed completely, as only parts of the PE were left at the sides and bottom. Therefore, the unburned PE reached the pyrolysis temperature and ignited. As the PE burned, high-temperature molten droplets were generated and stuck to the surface of the panel. Due to the internal heat

transfer to the unburned PE, the droplets caught fire and dripped to the bottom, resulting in a pool of burning, molten PE. Due to the combined effect of the burning panel and the molten PE, the fire spread.

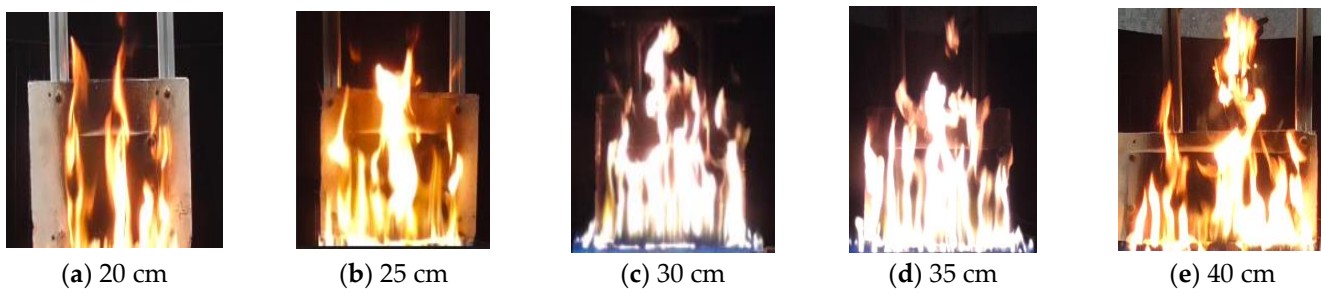

(**a**) 20 cm　　　　(**b**) 25 cm　　　　(**c**) 30 cm　　　　(**d**) 35 cm　　　　(**e**) 40 cm

**Figure 5.** Flame shapes of the metal sandwich panels for the different panel lengths in the stable combustion stage.

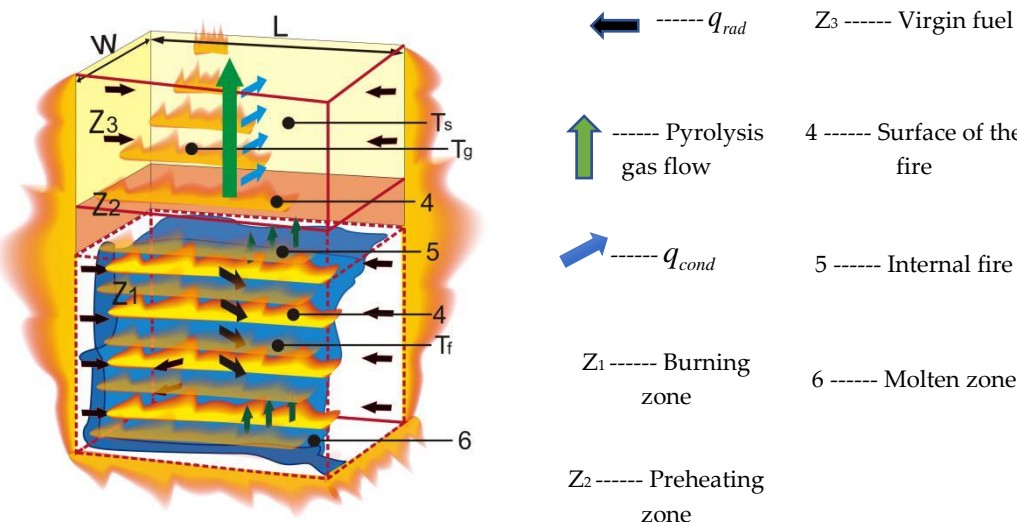

**Figure 6.** Diagram of the fire spread of the metal sandwich panel.

### 3.1.2. Bending Deformation and Top Flame

The part of the sandwich panel that was close to the fire source burned quickly, and the upper part shrunk due to heat. The flame spread rate was higher than the burnout speed of the sample. In addition, the bolts held the panel together during combustion, causing bending deformation of the aluminum alloy sheet (Figure 7), which accelerated the pyrolysis process, produced pyrolysis gases, and intensified the combustion of the sample. Bending deformation of the sandwich panel containing bio-derived constituents was also produced under the action of turbulent flame [25].

Due to the bending deformation, the combustion behavior differed on both sides of the panel with a length of 40 cm. The high-temperature area of the panel increased, which accelerated the pyrolysis process and produced large amounts of pyrolysis gases. The hot flammable gases rose and ignited externally, which caused the top flame phenomenon, as shown in Figure 8. Sheet deformation occurred between the sandwich panel and the insulation board. As the panel length increased, the sheet curvature increased, and vice versa, as shown in Figure 9. The bending deformation occurred because of the presence of the bolts on both sides of the panel. As the length increased to 40 cm, the combustion became more intense because of the top flame. The top flame also heated the top of the sandwich panel, which accelerated the pyrolysis process and caused greater deformation.

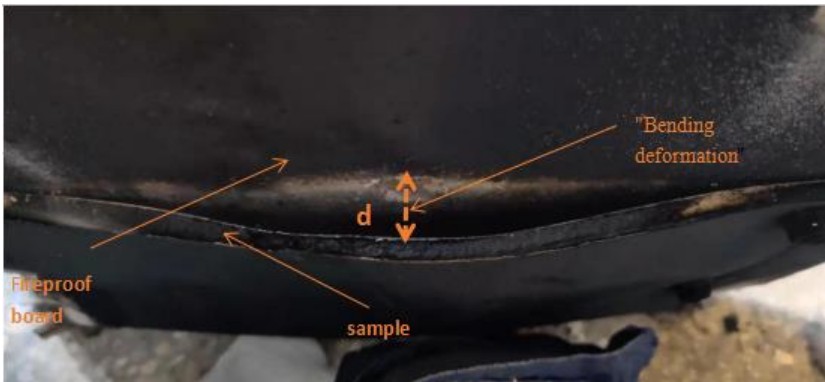

**Figure 7.** Bending deformation occurring during combustion.

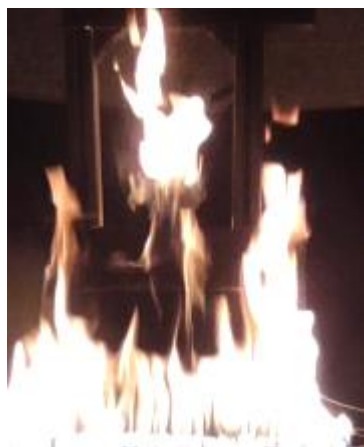

**Figure 8.** Top flame.

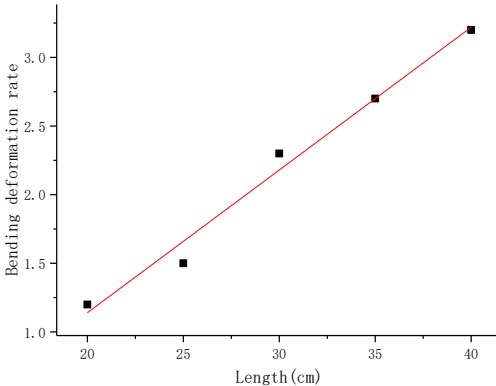

**Figure 9.** Bending deformation rate for the different panel lengths.

### 3.2. Dripping Behavior and Mass Loss of the Droplets

3.2.1. Dripping Behavior

During combustion, the melted PE expanded to the outside of the panel and burned due to the direct exposure to the fire. However, the hot, molten PE slowly flowed down and accumulated in the middle of the bottom part of the panel. When the gravity of the droplets exceeded the viscous force, the droplets fell into the bottom tray. In the preheating and initial stages of combustion, the droplets burned out promptly and could not maintain a fire [26]. However, in the stable stage, the droplets accumulated and fueled the fire, as shown in Figure 2. Because of the bolt, the droplets dripped from the middle of the sandwich panel. The drop time, drop position, and melt duration of the drops are shown in

Table 1. The high-temperature area and the area of the droplets increased with the length of the sandwich panel. In addition, due to the increasing of the heat transfer area, more PE was pyrolyzed, which increased the beginning of dripping time. The beginning of dripping time varied as a function of panel length.

**Table 1.** Drop time, drop position, and melt duration of the droplets under different working conditions.

| Number | Fire Source Power/kw | Sandwich Panel Length/cm | Dripping Position | Beginning of Dripping Time/s | Duration Time/s |
|--------|---------------------|--------------------------|-------------------|------------------------------|-----------------|
| $A_1$ | 5 | 20 | | $108 \pm 4$ | $1026 \pm 9$ |
| $A_2$ | 5 | 25 | | $142 \pm 7$ | $1132 \pm 11$ |
| $A_3$ | 5 | 30 | Middle of the bottom | $164 \pm 6$ | $1248 \pm 13$ |
| $A_4$ | 5 | 35 | | $178 \pm 11$ | $1260 \pm 17$ |
| $A_5$ | 5 | 40 | | $134 \pm 7$ | $1380 \pm 13$ |

### 3.2.2. Mass of the Droplets

The results of the mass changes in the droplets are shown in Figure 10. The mass of the droplets increased with an increase in the length of the sandwich panel. The droplet mass was lower for the panel with a length of 40 cm than that with a length of 35 cm, although the dripping frequency was higher. The mass of the droplets of the 40 cm long panel was lowest at the start of combustion. The mass change rate was higher because of the longer time interval of dripping for A1-A4 and the large mass of the droplets. The drip time of A5 was continuous and stable. The mass of the droplets was low, and the rate of mass change was low. Thus, the mass growth of A5 was a smooth curve. Although the mass growth of A5 was lower at the beginning, the accumulation rate was higher than that of A1-A4, and the mass growth was higher than that of the other four panels at 446 s. With the increase in length, the dripping time decreased, and the curve of the mass change increased gradually from the segmented line. Figure 3 shows that the fire extinguishing times of the burning droplets were different for different panel lengths, and the longest extinguishing time was observed for the panel with a length of 40 cm. The longer the burning time, the more unstable the mass of the droplets was. The mass of the droplets was not related to the mass growth of the droplets.

The average mass growth rates for the panel lengths of 20 cm, 25 cm, 30 cm, 35 cm, and 40 cm were 0.37 g/min, 0.67 g/min, 1.18 g/min, 1.77 g/min, and 2.56 g/min, respectively. The average mass loss rates were 0.03 g/min, 0.05 g/min, 0.1 g/min, 0.15 g/min, and 0.25 g/min, respectively. The average mass growth rate and loss rate of the droplets increased with the sample length.

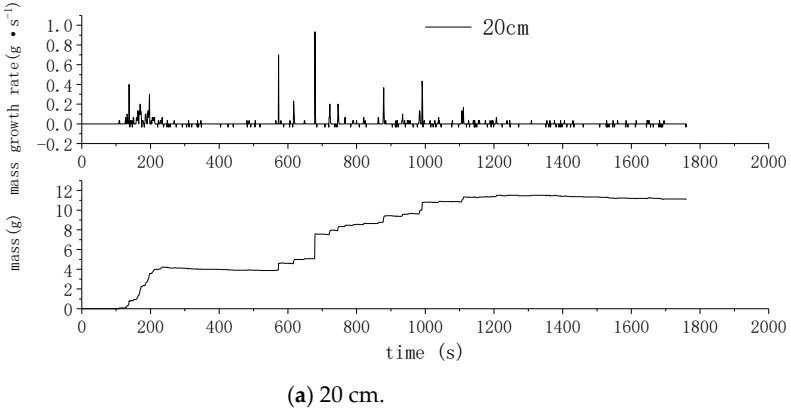

(**a**) 20 cm.

**Figure 10.** *Cont.*

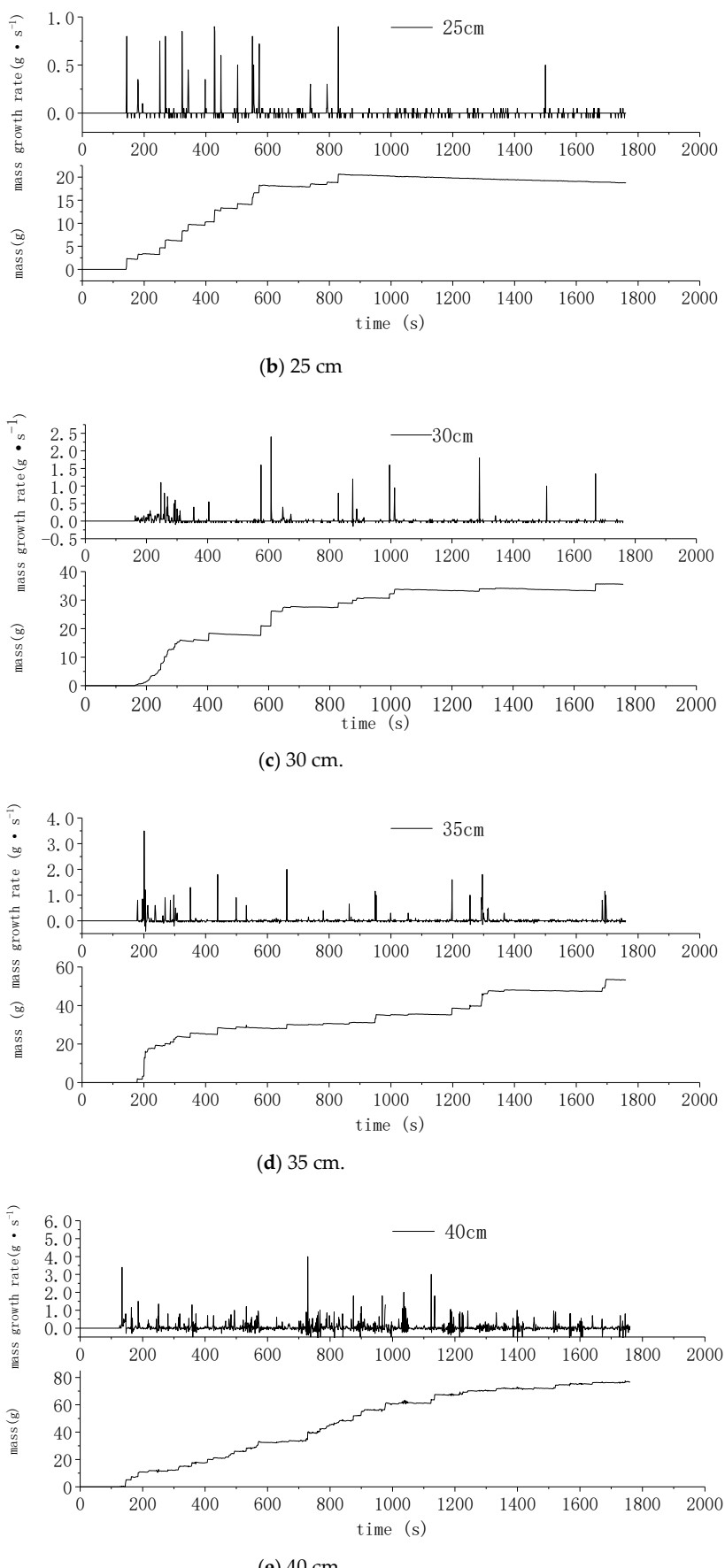

(**b**) 25 cm

(**c**) 30 cm.

(**d**) 35 cm.

(**e**) 40 cm.

**Figure 10.** Mass growth of the droplets for different panel lengths.

### 3.3. Temperature and Flame Spread Rate of the Sandwich Layer

The temperature curves are shown in Figure 11. The numbers 101, 102, 103, and 104 in the legend correspond to the thermocouple layouts $P_1$, $P_2$, $P_3$, and $P_4$ described in Section 2.1.4, respectively. It was observed that the temperature increased with the combustion time. The peaks in the curves indicate the time when the thermocouples were exposed to the fire. The trend of the temperature change was the same at all points. An increase in the temperature was observed for the 30 cm sample at about 1100 s, and the times were lower for the 35 cm and 40 cm samples. The discussion of the droplet mass in the previous section shows that the mass loss increases as the sample length increases. A larger droplet mass means that the un-pyrolyzed PE material is more likely to come into direct contact with the flame and heat up rapidly. The highest temperature occurred at $P_1$. The maximum temperatures under the five conditions were 849.806 °C (1650 s), 860.943 °C (1800 s), 926.746 °C (1700 s), 952.463 °C (1604 s), and 967.332 °C (1611 s), respectively. The maximum temperature at $P_1$ increased with an increase in the sample length. The rate of change of the temperature increased as the length increased. At longer lengths, there were more peaks, and the difference between the peaks and troughs was larger. This result was attributed to the increases in the flame pulsation with the increasing sample length, which was caused by the thermal buoyancy and instability of the boundary layer. There was a positive correlation between the intensity of flame pulsation and the instability of buoyancy. The Grashof number ($Gr$) was used to represent the instability of buoyancy [4].

$$Gr = g\beta(T_f - T_\infty)L^3\sigma_g{}^2/\mu^2 \tag{1}$$

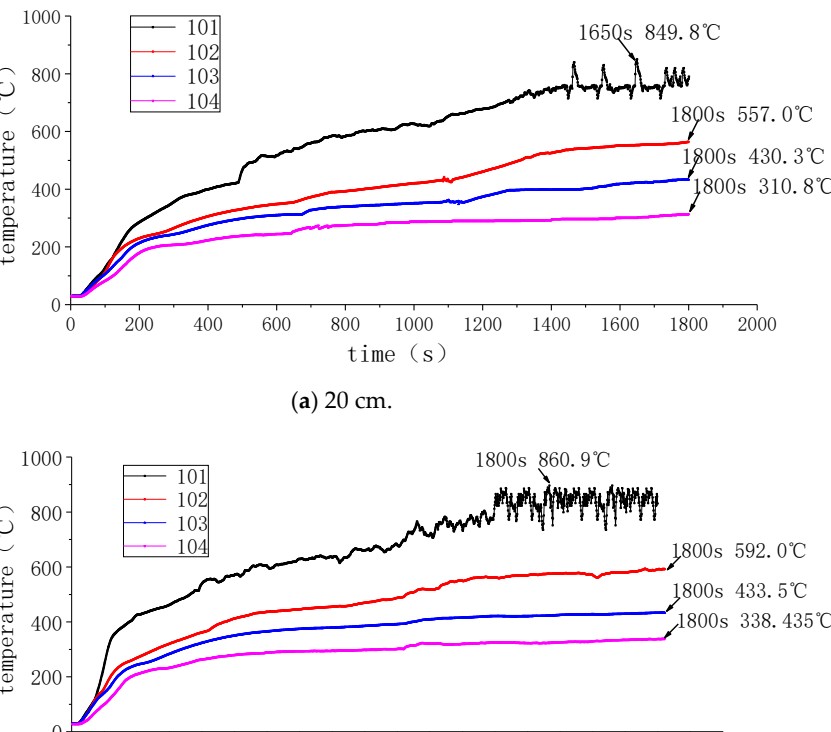

(**a**) 20 cm.

(**b**) 25 cm.

**Figure 11.** *Cont.*

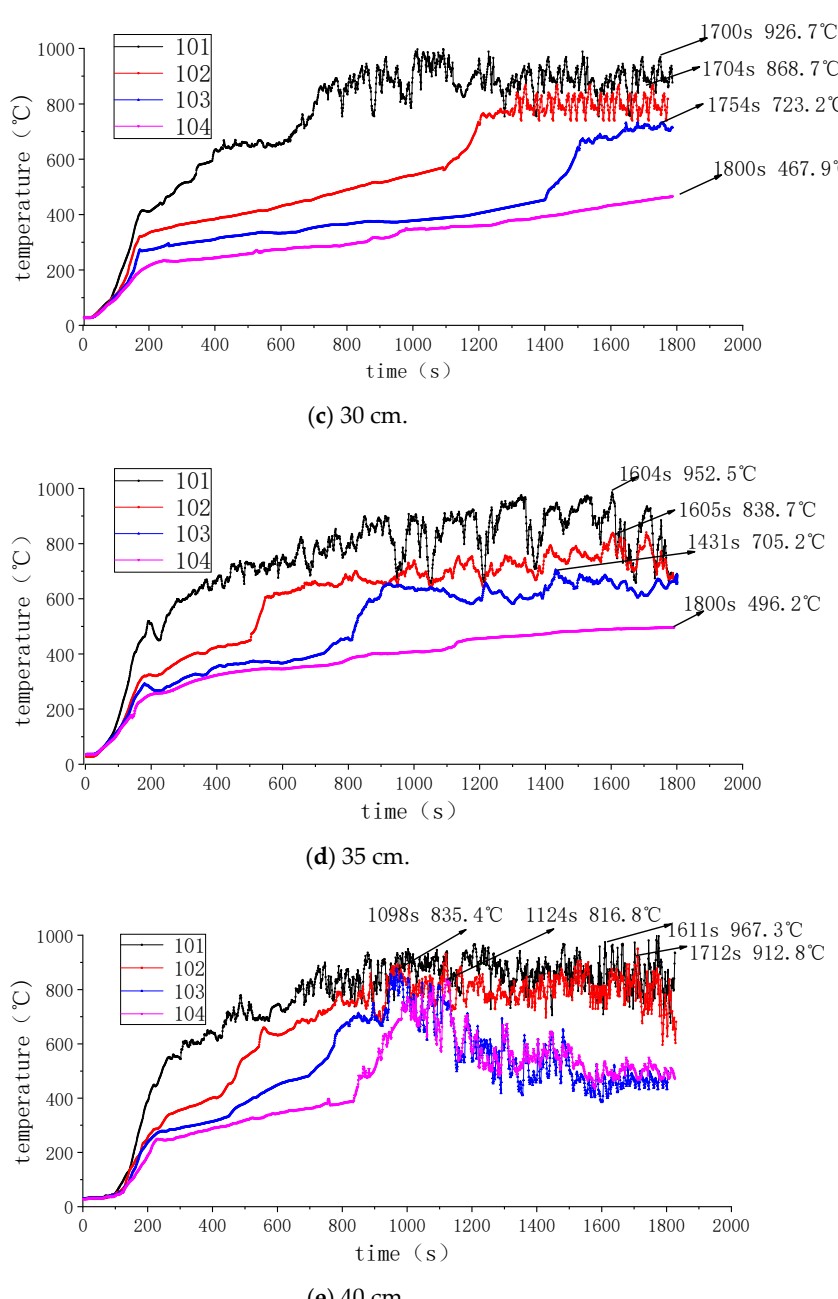

**Figure 11.** Temperate of the sandwich layer under different conditions.

The length of the sample was set as the characteristic length (*L*). $\sigma_g = pM/RT$ in Equation (1). In general, the Grashof number (*Gr*) is employed to represent the instability of buoyancy. The degree of flame pulsation frequency can be indirectly derived from the Grashof number. There was a positive correlation between the *Gr* and *L*. The larger the length, the worse the stability of the buoyancy was, and the higher the frequency of the flame pulsation was, which caused higher peaks and greater difference during combustion, as shown in Figure 11. Although melting and burning occurred in multiple locations under the same condition, the magnitude of the temperature change decreased as the distance from the fire source increased [27,28]. The four stages of the combustion described in Section 3.1.1 are clearly shown. Figure 11 also shows that the flame spread to the top of the panels in the different samples. The time between two adjacent points under the same conditions decreased. As the length increased, the intensity of the combustion of the sandwich panel increased, and more heat was transferred to the panel and the PE.

The temperature of the 40 cm-long panel was the highest, and the time between the two adjacent points was the smallest. However, the influence of the top flame on $P_3$ and $P_4$ was greater than that of the fire source, which was the reason why the temperature was higher at $P_4$ than at $P_3$.

### 3.4. Mass-Loss Rate of the Metal Sandwich Panel

The mass-loss rate of the sandwich panel reflected the severity of the combustion and the fire intensity. As shown in Figure 12, the mass decreased continuously, and the rate of mass loss first increased, stabilized, and then decreased.

The average burning rate of the sample during the steady combustion phase was calculated. The results showed that the rate of mass loss of the metal sandwich panel increased with an increase in the panel length. The average mass loss rate $m_a$ was deduced using the methodology described by the authors of Ref. [29]. $m_l = m_a / L$ is defined as the average mass loss rate per unit length. The larger the $m_l$ value, the more intense the combustion is, and the greater the fire risk is. The results are shown in Figure 13.

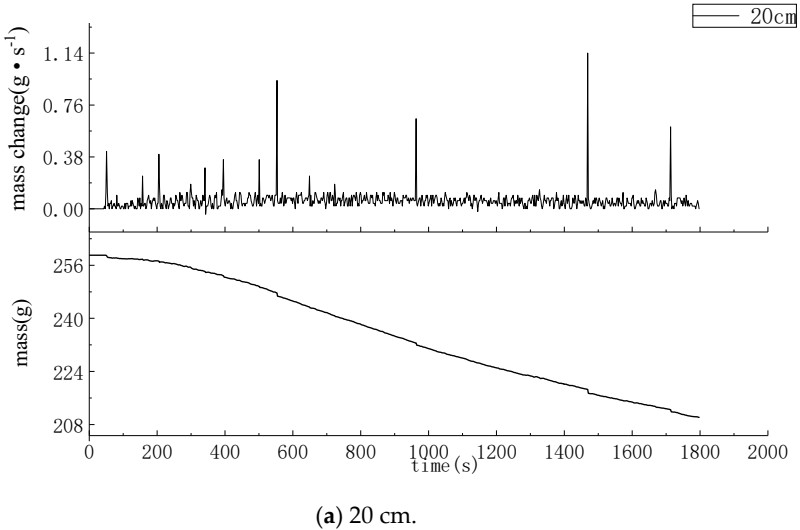

(**a**) 20 cm.

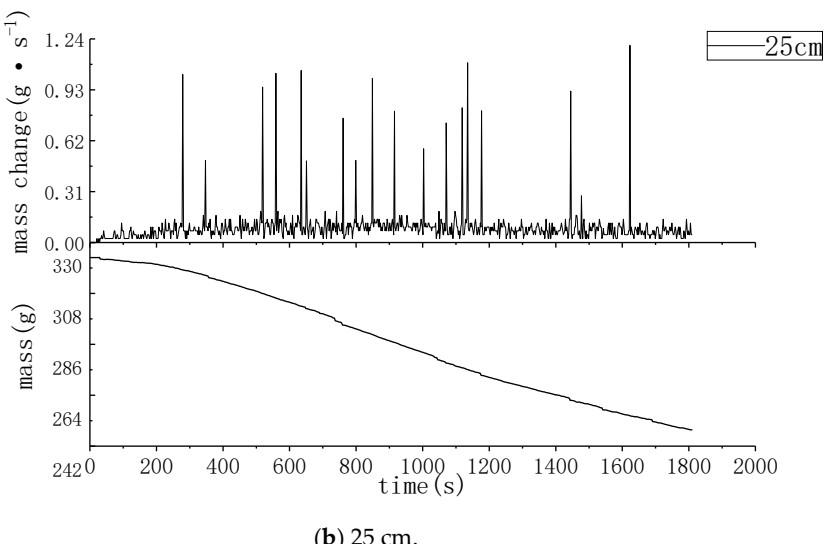

(**b**) 25 cm.

**Figure 12.** *Cont.*

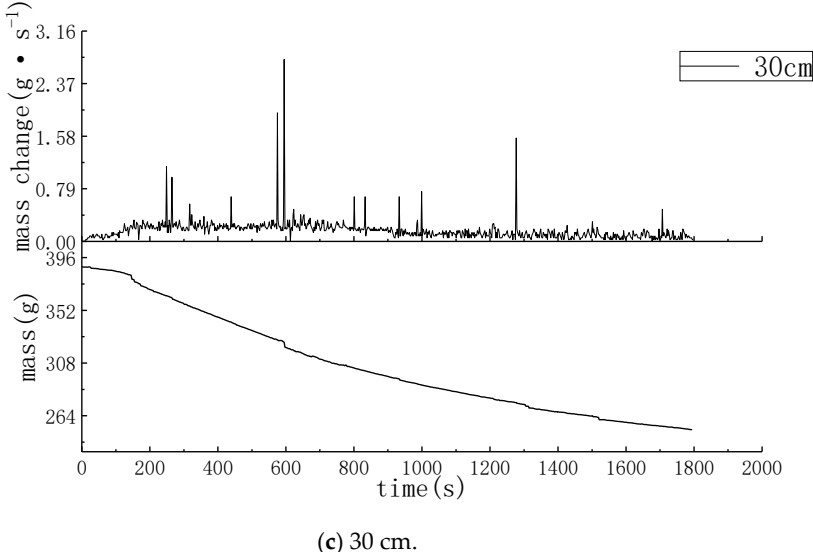

(**c**) 30 cm.

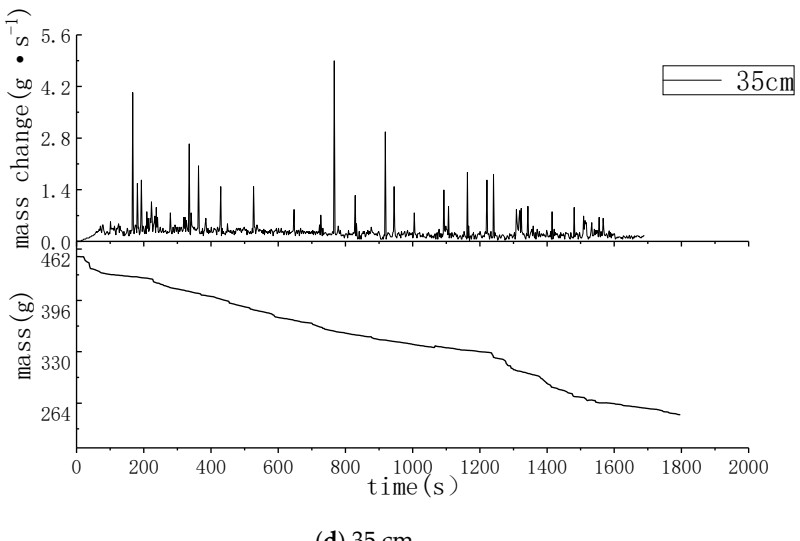

(**d**) 35 cm.

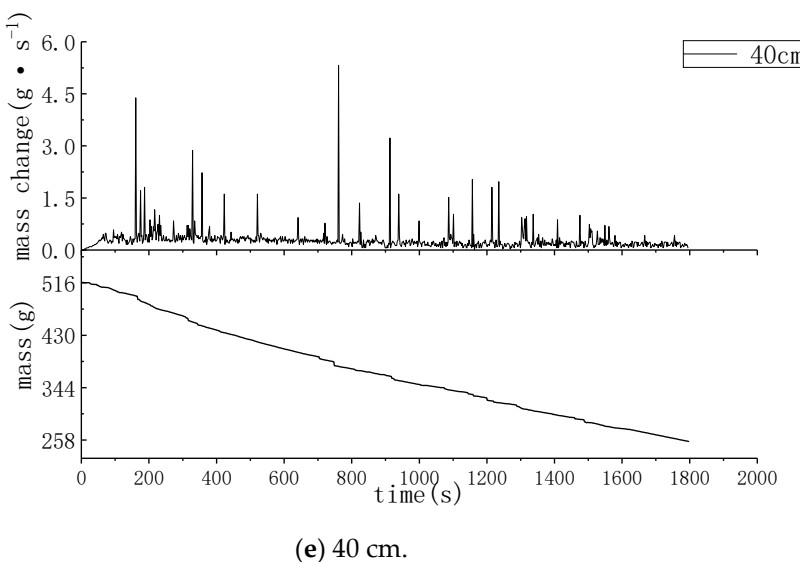

(**e**) 40 cm.

**Figure 12.** Mass changes of the panels with different lengths.

As shown in Figure 13, $m_l$ increased with an increase in the sample length. The exception was the 40 cm-long sample. The mass-loss rate was determined by the heat flux. The heat flux consists of radiant, convective, and solid-phase heat flux. Since the layer was covered by the aluminum laminate, there was no gap between the PE core and the aluminum laminate, and heat convection did not occur. The convective heat flux was ignored. However, because of the heat conductivity of the metal, the solid-phase heat flux had a significant influence on the mass loss rate of the sandwich panel. The flame flux consisted of solid-phase heat flux and radiant heat flux [30].

$$m_a = C(q_{cond} + q_{rad}) \tag{2}$$

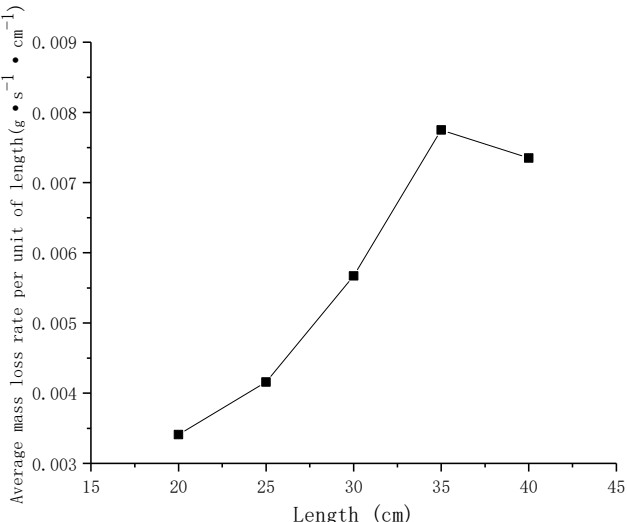

**Figure 13.** Changes in the average mass loss rate per unit length of the panel.

The whole heat conduction process can be regarded as bottom-up. The heat flux in the thickness direction and length direction of the sandwich panel is relatively weak, so the whole heat conduction process can be simplified to a vertical transient heat transfer process. Therefore, the heat flux density was obtained according to Fourier's law [24]:

$$q_{cond} = k_x(\partial T / \partial x) \tag{3}$$

The radiant heat flux was obtained using Equations (4) and (5):

$$q_{rad} = \varepsilon_f \sigma (T_f^4 - T_p^4) \tag{4}$$

$$\varepsilon_f = 1 - \exp(-k_s L) \tag{5}$$

$m_l$ was obtained by Equations (2)–(5):

$$m_l = Lk_x(\partial T / \partial x) + \sigma L(T_f{}^4 - T_p{}^4)(1 - \exp(-k_s L)) \tag{6}$$

where $C$ is a constant, and $L$ is the characteristic length in this study. Equation $m_l = m_a / L$ indicates a positive correlation between the average mass loss rate per unit length and the characteristic length. The same conclusion in the XPS sandwich panel in high-rise building fire spreading was also produced [31]. It was deduced that $m_l$ rose with an increase in the sample length. The combustion in the stable phase increased in intensity with an increase in the sample length. The flame was generated by the diffusing pyrolyzed gases that moved toward the boundary region of the sheet, which increased the heat radiation and heat conduction from the surface of the aluminum panel. The surface temperature of the sample increased. The insulation performance of the PE core was high, and the thermal conductivity was low. The $\partial T$ between the aluminum layer of the sandwich panel and the

PE core increased with an increase in the sample length, and $m_l$ also increased. For the 30 cm sample, the frequency of the large, melted droplets was significantly lower than that of the 25 cm sample. Therefore, the higher integrity of the 30 cm sample during the combustion resulted in a longer stable combustion stage. However, a significantly higher number of droplets were observed in the 40 cm-long sample than the other samples. More heat was removed from the combustion zone, which decreased the heat transfer from the flame and resulted in a low $m_l$.

### 3.5. Flame Spread Rate

The flame spread rate is a crucial parameter to determine the combustion characteristics of metal sandwich panels and assess the fire risk. The purpose was to measure the flame propagation speed in the vertical direction in the middle of the sandwich panel. When the flame spread to the thermocouple, the temperature rose rapidly, and the temperature fluctuated. Figure 14 shows the flame spread rate. It was observed that the flame spread rate increased with an increase in the sample length. Heat feedback was the decisive factor in the speed of the fire spread. Heat feedback increased with an increase in the sample length, which caused the flame spread rate to increase as the length increased [32].

As shown in Figure 14, the fire spread rate increased from $P_1$ to $P_2$. However, the fire spread rate from $P_2$ to $P_3$ began to decrease. The fire source and the combustion of the PE core were the main factors affecting the flame spread rate and were crucial parameters in the different stages. $P_1$ and $P_2$ were close to the source of the fire, which played a dominant role in influencing the fire spread. $P_3$ and $P_4$ were far from the fire source, and the combustion of the PE significantly influenced the fire spread rate. The combustion of the PE had a negligible effect on the flame spread rate due to the aluminum alloy layer. The fire spread rate of the 40 cm sample increased from $P_3$ to $P_4$. When the flame spread to $P_3$, the PE core on both sides was burnt out by the side flames. There was little insulation material left around $P_3$ to $P_4$ in the vertical direction. The combustion of the PE insulation material also substantially affected the flame spread rate.

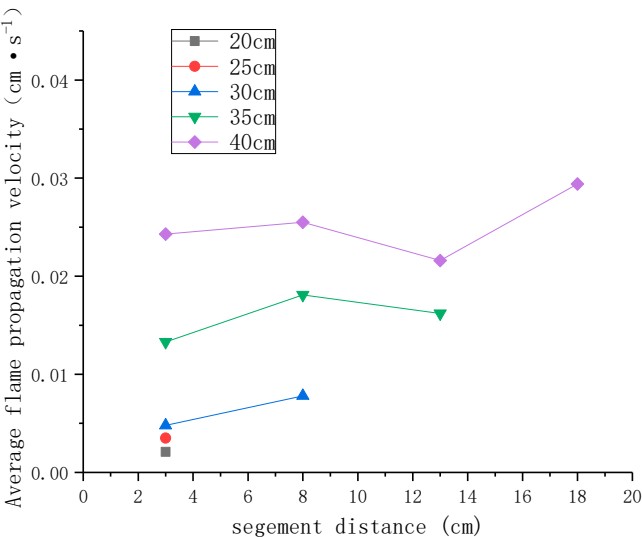

**Figure 14.** Fire spread rate under different conditions.

### 3.6. Average Flame Height

The flame height is a vital characteristic in the analysis of the flame spread. The OTSU algorithm was used to convert and obtain the flame height. Since flame height fluctuates over time, the height of the flame with a probability of 0.5 was defined as the average flame height of the sandwich panel [33]. The results are shown in Figure 15.

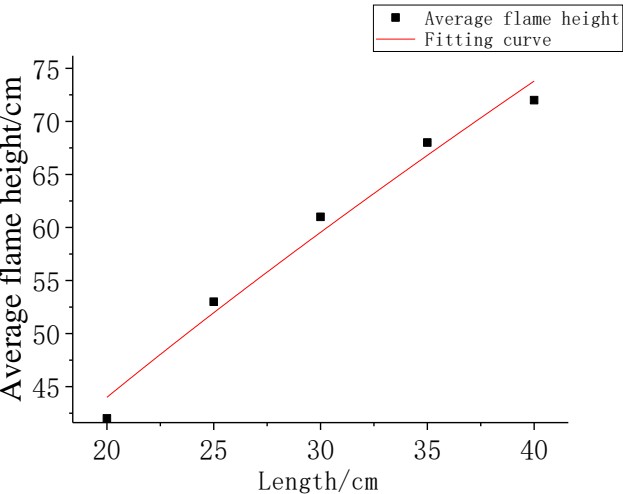

**Figure 15.** Average flame height versus the sample thickness under different conditions.

In order to establish the connection between the reduced size and the full size, the important physical parameters must satisfy the Froude similarity model. The Froude similarity criterion is to satisfy the kinetic similarity to ensure the Froude number of the reduced size and the full size are the same. The Froude number characterizes the relationship between the inertial force and the buoyancy ratio of the fluid. The magnitude of flame height depends on $Fr$. This experiment can take a dimensionless flame height, explore its linear relationship with $Fr^n$, and derive the relationship between the average flame height and length. $Fr$ represents the ratio of the inertial force of the fluid to the heavy force level (the buoyancy force in this experiment) [34]:

$$Fr^n = \mu_0^2 / (Lg) \tag{7}$$

where $H/L$ is the dimensionless flame height,

$$H/L = Fr^n = (\mu_0^2 / Lg) \tag{8}$$

$$H/L = L^{-n} \tag{9}$$

The result of this experiment was $n = 0.75$.

The average flame height and sample length agree with the following equation:

$$H = 4.71 \times L^{0.75} \tag{10}$$

## 4. Conclusions

In this study, the fire spread rate and droplet characteristics of metal-PE sandwich panels were investigated during combustion. The droplet characteristics, mass loss, flame temperature, average flame height, and flame spread rate of different lengths of sandwich panels were studied. The results provide a theoretical basis for comprehensive hazard assessment and fire prevention methods for sandwich panels. A schematic diagram of the upward fire spread of the metal sandwich panel was established. The results of this work are summarized as follows.

(1) During the combustion of the sandwich panel, the PE core melted, and droplets were formed. The droplet characteristics varied under the different conditions and in different stages of the combustion process. As the panel length increased, the mass of the droplets, the area of the molten droplets, the average mass loss rate, and the average mass growth rate increased. Due to the presence of bolts attached to the sandwich panel, bending deformation occurred. The level of bending deformation increased with an increase in the sample length. Since the pyrolysis gases rose and ignited above the 40 cm-long sample, the top flame phenomenon occurred.

(2) The mass loss of the metal sandwich panel increased, stabilized, and then decreased. As the sample length increased, $m_l$ increased. A significantly larger number of droplets were observed in the 40 cm-long sample than the other samples, and heat and fuel were removed from the combustion zone. As a result, the flame weakened, and the heat transfer decreased, causing a low $m_l$.

(3) The maximum temperature of the panels with different lengths occurred at point $P_1$. The maximum temperature at all points and the pulsation frequency of the flame increased for a longer sample length. More peaks and larger differences between the peaks and troughs were observed with the increasing sample length. The fire only spread to $P_1$ in the samples with lengths of 20 cm and 25 cm, whereas it spread to $P_3$ in the samples with lengths of 30 cm and 35 cm and to $P_4$ in the 40 cm-long sample. Because of the top flame phenomenon, the temperature of $P_4$ during combustion was higher than that of $P_3$.

(4) The average flame spread rate increased with the increasing sample length. The fire spread rate increased from $P_1$ to $P_2$ and decreased from $P_2$ to $P_3$ for the 30 cm and 35 cm samples. In the 40 cm sample, the average fire spread rate increased from $P_1$ to $P_2$, decreased from $P_2$ to $P_3$, and increased from $P_3$ to $P_4$.

(5) The average flame height increased as the length of the metal sandwich panel increased. The relationship between $H$ and $L$ was $H = 4.71 \times L^{0.75}$.

**Author Contributions:** Conceptualization, R.Z.; methodology, R.Z. and Y.F.; formal analysis, R.Z. and Z.C.; investigation, J.Q. and Y.F.; writing—original draft preparation, R.Z. and Y.F.; writing—review and editing, R.Z., Z.C., Z.Y. and J.Q.; supervision, J.J. All authors have read and agreed to the published version of the manuscript.

**Funding:** This research was funded by the National Natural Science Foundation of China (Grant No.51874183, 51874182), Six Talent Peaks Project of Jiangsu Province (Grant No. 2019-JZ-017), Key Laboratory of Fire Protection and Rescue of Ministry of Public Security (Grant No. KF201810) and Postgraduate Research & Practice Innovation Program of Jiangsu Province (Grant No. SJCX20_0394).

**Institutional Review Board Statement:** Not applicable.

**Informed Consent Statement:** Not applicable.

**Data Availability Statement:** Not applicable.

**Acknowledgments:** This research was funded by the National Natural Science Foundation of China (Grant No.51874183, 51874182), Six Talent Peaks Project of Jiangsu Province (Grant No.2019-JZ-017), and Key Laboratory of Fire Protection and Rescue of Ministry of Public Security (Grant No. KF201810) and Postgraduate Research & Practice Innovation Program of Jiangsu Province (Grant No. SJCX20_0394).

**Conflicts of Interest:** The authors declare that they have no known competing financial interests or personal relationships that could have appeared to influence the work reported in this paper.

## Abbreviations

| | |
|---|---|
| $B$ | Combustion efficiency |
| $C$ | Constants |
| $Fr$ | Froude number |
| $g$ | Acceleration of gravity (m s$^{-2}$) |
| $Gr$ | Grashof number |
| $\Delta H_c$ | Heat of complete combustion (kJ s$^{-1}$) |
| $\Delta H_{eff}$ | Effective heat of combustion (kJ s$^{-1}$) |
| $k_x$ | Thermal conductivity (Js$^{-1}$ m$^{-1}$ k$^{-1}$) |
| $k_s$ | Absorption coefficient of carbon particles |
| $L$ | Characteristic length of the sample (cm) |
| $m_1$ | Gaseous mass (g s$^{-1}$) |
| $m_a$ | Average mass loss rate (g s$^{-1}$) |
| $m_l$ | Average mass loss per unit of length (g s$^{-1}$ cm$^{-1}$) |
| $m'_n$ | The mass loss rate at the n$^{th}$ second (g s$^{-1}$)M Quality (kg) |

| | |
|---|---|
| $p$ | Pressure (kPa) |
| $q_{cond}$ | Solid-phase heat flux (W m$^{-2}$) |
| $q_{rad}$ | Radiative heat flux (W m$^{-2}$) |
| $R$ | Gas constants |
| $T$ | Thermodynamic temperature(K) |
| $T_f$ | Flame temperature (K) |
| $T_p$ | Pyrolysis temperature (K) |
| $T_\infty$ | Ambient temperature (K) |
| $T_g$ | Gas temperature (K) |
| $T_s$ | Solid surface temperature (K) |
| $\varepsilon_f$ | Flame emissivity |
| $\sigma$ | Stefan–Boltzmann constant |
| $\beta$ | Volume thermal expansion coefficient |
| $\mu$ | Coefficient of motion viscosity |
| $\mu_0$ | Fluid motion speed |

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
