# Peer review of "Fire Spread Characteristics of Metal-Polyethylene Sandwich Panels"

_buildings, doi:10.3390/buildings11090396_

Round 1
Reviewer 1 Report
The manuscript entitled “Fire spread characteristics of metal-polyethylene sandwich panels” meets the scope of Buildings. It has a certain significance for the practical application in understanding the behavior of sandwich panels when exposed to fire. The manuscript is well written with some minor mistakes. Moreover, few and rather bold explanations are given about the results and about the equations which were used. Thus, I recommend major revisions before accepting the work. Some comments are as follows:
- In the Introduction, authors cite various works which evaluated the fire behavior of sandwich panels, but no critical point of view is given.
- What is EPX on line 60, page 2?
- Why to set the power of the fire source to 3.33 kW? By the way, on Table 1, the fire power is given as 5 kW.
- On section 3.1.1, authors classified the combustion phenomena into four stages. However, it is not very clear, for example, the difference between the “initial stage of combustion” and the “stable stage of combustion” seems to be classified only by feeling. Do authors measured the time interval between droplets in order to establish reproductible measurements?
- In figure 5, the shape of the flame as a function of panel length is given. Moreover, the shape is also function, for example, of time. Are this images representative? Authors must give more details about this image.
- Standard deviation values should be given in Table 1.
- From table 1, it can be seen that the beginning of dripping time varies as a function of panel length. Moreover, this difference is greater between 20 and 25 cm, but the variation of lower from 25 to 40 cm. Do authors have any explanation?
- How authors explain the observed differences on the mass rate changes of droplets and its mass (section 3.2.2)?
- Legends of Figure 11 must be reviewed. What means 101, 102 and so on? P1, P2…? Which is the significance of 3 decimal values?
- On Figure 11, it is possible to observe an increase in temperature for the sample with 30 cm at about 1100 s, and at lower times for the 35 and 40 cm samples. Do authors have some explanation?
- On sections 3.4, 3.5 and 3.6 some equations are shown, but no details about the use of them nor the hypothesis used. Some bold conclusions are given with no values. Authors should improve these sections with more details and values that supports their conclusions. For example, why could you use the Grashof number in this case? The solid phase heat flux was deduced using the equation 4, but no detail about the hypothesis is given. Some more explanations about the use of Froude number and its importance for the present work should be given.
- On section 3.6, which algorithm was used to obtain the flame height?
- Not all the symbols are listed in Appendix “List of symbols”.
Author Response
We appreciate for Reviewers’ warm work earnestly。
Please see the attachment.

Reviewer 2 Report
The authors built nice device to study fire spread in metal-polymer foam panels. But unfortunately, the paper in general is very confusing.
- Different length panels from 20 to 40 cm were studied. Because flame spreads much faster in vertical direction one can expect that panels will be mounted with long side being in vertical position. But it was not the case. It was never described in the experimental part. You really need to pay attention to Figure 5, to understand that the panels were mounted horizontally and ignition source was not always in the center.
- Experimental time cut off as shown in Figs. 10-12 was 1800 sec (30 min). But the weight didn't stabilize after 30 min (Fig. 12). Why? Does it mean that burning continued but recording was cut prematurely? Then all discussion that flame never reached top of the panels for 20-35 cm long panels (Fig. 11) is wrong. It was simply not given enough time to get there.
- Maximum flame temperature in Fig. 11 was picked up as an arbitrary peak temperature recorded by thermocouple. It is reported with enormous precision of 3 decimal numbers. Doesn't make sense, especially with significant signal fluctuations. It makes more sense to report average temperature.
- Flame spread diagram in Fig. 6 requires significant explanation. I could not understand it.
Author Response
We appreciate for Reviewers’ warm work earnestly.
Please see the attachment.

Reviewer 3 Report
Thank you for your contribution to science. Please incorporate the following comments and suggestions to your article:
- Specify if sandwich panels were used in the construction of Busan Industrial Park in South Korea and in Grenfell Tower and if they contributed to the severity of the fires. If not, provide other examples of industrial fires were the panels played a significant role.
- Provide a more detailed description of Figure 6
- Define the thickness of your samples (it is not mentioned in the text and the schemes do not correspond with the pictures)
- What was the power output of the fire source? Line 124 - 3.33 kW, table 1 - 5 kW
- Figure 12: double check units in the first graph ( mass(s-1) )
- Equation 2 is unnecessary (arithmetic mean is a well-known fact)
- Line 289: typo in the word "thee"
- Figure 13: explain the units used in the graph
- Figure 14: typo in the word " segement"
- There is no discussion of the results with other researchers dealing with similar topic in the article.
Author Response

(The authors gave the same response as above.)

Round 2
Reviewer 1 Report
The authors made some changes compared to the first version of their manuscript i order to answer the questions. I recommend now to accept the manuscript in its current version.
Author Response
Dear Reviewer:
In this round, we do not need to revise the manuscript.
Thanks for your help and your valuable comments.
Best Regards.
Ru Zhou
Reviewer 2 Report
Accept with proposed corrections and additions.
Author Response

(The authors gave the same response as above.)
